

# Polarization-sensitive imaging in magnetic environments

**Nejc Blaznik⋆, Dries van Oosten and Peter van der Straten**

Debye Institute for Nanomaterials Science and Center for Extreme Matter
and Emergent Phenomena, Utrecht University

⋆ n.blaznik@uu.nl

## Abstract

Nondestructive spin-resolved imaging of ultracold atomic gases requires calculating the differences of the refractive indices seen by two circular probe polarizations. Perfect overlap of the two images, corresponding to two different polarizations, is required well below the feature size of interest. In this paper, we demonstrate that the birefringence of atoms in magnetic field gradients results in polarization-dependent aberrations in the image, which deteriorates the overlap. To that end, we develop a model that couples atomic tensor polarizability with position-dependent spin orientation and yields aberration predictions for accumulated phase shifts in arbitrary field geometries. Applied to data from an ultra-cold atomic cloud trapped in a Ioffe-Pritchard trap, the model quantitatively reproduces the observed distortion across a range of temperatures. A residual offset of $\sim 1\,\mu$m remains even under uniform field conditions, likely due to optical asymmetries. For images obtained through off-axis holography, the full complex field of the probe enables post-processing removal of all magnetically induced aberrations through a single numerically calculated Fourier-space phase mask.

| | |
|---|---|
| Received | 2025-07-24 |
| Accepted | 2025-10-06 |
| Published | 2025-10-31 |

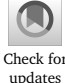

# 1   Introduction

Accurate mapping of atom distributions across spin states in ultra-cold ensembles underpins precision studies of spinor Bose-Einstein condensates (BEC), quantum magnetometry, and quantum simulation [1–3]. The properties of the various Zeeman sublevels are mainly studied in one of two ways; spatial separation via Stern-Gerlach (SG) deflection during time-of-flight, and polarization-sensitive dispersive imaging. Stern-Gerlach analysis relies on applying a short magnetic-field gradient after release from the trap, causing different spin components to become spatially separated, such that a single absorption image yields direct information about each substate population. The SG technique has been used to investigate the dynamics and phase transitions in cold spinor gases [4–6] as well as to study spin domain formation and relaxation [4,7–10] and to investigate magnetic solitons [11,12]. However, the method requires ballistic expansion that destroys the sample, integrates out *in-situ* correlations, and demands magnetic field gradients introduced by large currents that can lead to spurious forces.

In contrast, polarization-based methods are minimally invasive and exploit state-dependent light-matter interactions. These methods include circular-polarization absorption imaging [13, 14], dispersive Faraday rotation [15, 16], ellipticity measurements with a linearly polarized probe [17], phase-contrast and dark-ground imaging, and more recently, spin-dependent off-axis holography (SOAH) [18, 19]. Each of these techniques relies on the fact that atomic spins couple with different strengths to two probe beams of different circular polarizations. This relative phase shift between the two polarizations allows for the extraction of spin-dependent densities. Optically, imaging is typically performed using a single linearly polarized probe beam, which is decomposed into components $\sigma^+$ and $\sigma^-$, either in the setup using a combination of waveplates or by decomposition of a linear to circular basis during the analysis.

In recent experiments on spin-dynamics using SOAH [19], we observed a systematic $1.5\,\mu$m displacement between the reconstructed $\sigma^+$ and $\sigma^-$ images. We trace this shift to weak birefringence in the imaging optics. However, its presence led us to examine a much greater effect caused by magnetic field gradients. In the present article, we report on a comprehensive analysis of this effect. We show that position-dependent variations of the local magnetic field imprint systematic distortions in reconstructed profiles, presenting a fundamental limitation to all methods that extract local spin-density contrast by computing the pixel-by-pixel ratio of $\sigma^+$ and $\sigma^-$ images. As the magnitude of such a shift exceeds the typical spin-domain sizes present in the condensates, all such methods introduce significant artifacts that require post-processing correction.

In this paper, we develop a general model of spin-dependent birefringence in realistic magnetic environments by combining the tensor-polarizability formalism with local quantization-axis rotations, allowing for a position-dependent description of the phase shifts for any input polarization and field configuration. To model the distortions caused by the magnetic fields, the tensor polarizability is derived as a function of the angle between the spin quantization axis and the light propagation direction. Next, a three-dimensional map of the magnetic field vector is constructed based on the trap coil geometry. Finally, the corresponding Euler angle rotations are applied to the atomic density matrix, from which the final expressions for the column-integrated phase delay are derived.

We validate these predictions using SOAH, by studying temperature-dependent distortions of the imaged column-density profiles of the atoms trapped in a Ioffe-Pritchard trap. Understanding the impact of magnetic environments on polarization-sensitive imaging methods is essential to refining their use in future experiments and can be used to more reliably and accurately interpret spin-density measurements.

## 2 Experimental methods

To experimentally probe the effects of magnetic fields, we trap and cool sodium atoms in a magnetic trap and monitor them during the final stage of evaporative cooling. Using spin-dependent off-axis holography (SOAH), we image both the density and spin distribution of the cloud through the last 25 seconds of cooling, to and past the Bose-Einstein condensation threshold. Because SOAH records the full optical field, we can choose the probe detuning such that absorptive effects that lead to heating are suppressed by a factor of $\sim 1300$, while the phase signal remains appreciable. This makes it possible to follow a single cloud throughout the experiment without significant particle loss, capturing its evolution across the transition and revealing systematic shifts between the spin-resolved images.

A cold cloud of about $3 \times 10^8$ $^{23}$Na atoms is cooled to about $T \approx 800$ $n$K (which is below the critical temperature for BEC under our experimental conditions) using laser cooling and forced evaporative cooling. Atoms are held in an Ioffe-Pritchard trap using a clover-leaf coil geometry for the quadrupole field and two pairs of Helmholtz coils for the curvature and antibias fields [20, 21]. At a radial gradient of $B' = 11.4\,\mathrm{G\,mm^{-1}}$, an axial curvature of $B'' = 0.347\,\mathrm{G\,mm^{-2}}$, and a uniform bias field of $B_0 = 2.0\,\mathrm{G}$, the radial and axial trap frequencies are approximately $(\omega_r, \omega_z) = 2\pi \times (140, 15)\,\mathrm{Hz}$, respectively. Only the atoms in the Zeeman sublevel $|F = 1, m_F = -1\rangle$, with spins anti-aligned with the magnetic field quantization axis, remain trapped.[1]

During the last 25 seconds of cooling, spin-resolved imaging is performed by employing spin-dependent off-axis holography (SOAH) [19]. In this method the linearly polarized probe beam interacts with the atomic cloud, such that the two circular components acquire different phase shifts depending on the local spin state. The probe beam is then interfered on the camera with two circularly polarized reference beams. Each of the two reference beams is incident at a different, small angle, which produces spatial interference fringes for both $\sigma^+$ and $\sigma^-$ beams on the camera in a single exposure. As the two reference beams hit the camera under different angles, the hologram can be Fourier filtered in such a way, to reconstruct the entire complex optical field for each polarization component separately.

In our implementation, the probe beam is detuned by $\delta = -175$ MHz (about $\delta/\Gamma \approx 18$ linewidths) from the $|F = 1\rangle \rightarrow |F' = 1\rangle$ resonance. As the absorption scales approximately with $(\Gamma/2\delta)^2$, while the phase delay scales with $\Gamma/2\delta$, the relative strength of phase delay compared to absorption is enhanced by a factor of $2\delta/\Gamma \approx 36$. Thus, the effective OD is strongly suppressed, making the imaging minimally destructive, while maintaining good sensitivity to spin-dependent phase shifts.

In total, 50 images are taken of the atoms at a rate of 2 Hz, with probe pulses 25 $\mu$s long with an intensity of roughly 80 $\mu$W/cm$^2$, corresponding to a saturation parameter of $s \approx 1.0 \times 10^{-5}$. This ensures that the imaging is done well within the linear dispersive regime and does not cause significant additional heating. The final stage of cooling reduces the temperature from 20 $\mu$K to below 1 $\mu$K with the onset of condensation after 20 seconds at a temperature of $\sim 1.03$ $\mu$K. Following the Fourier-cut analysis described in Ref. [19], a pair of images corresponding to opposite polarizations ($\sigma^+$, $\sigma^-$) are reconstructed. Examples of such images are shown in Figure 1a, b.

Each image is fitted with a two-dimensional density profile from which physical variables such as temperature, particle number, and chemical potential are obtained, as well as the radial and axial centers of the distributions. In Figure 1e the radial centers are plotted as a function of time as the cloud is cooled below the transition temperature. We observe a clear shift between

---

[1]It is worth noting that although all the atoms are in the same orientation relative to the magnetic field, the magnetic field lines are spatially varying with respect to the laboratory frame, and thus the $k$-vector of the probe beam.

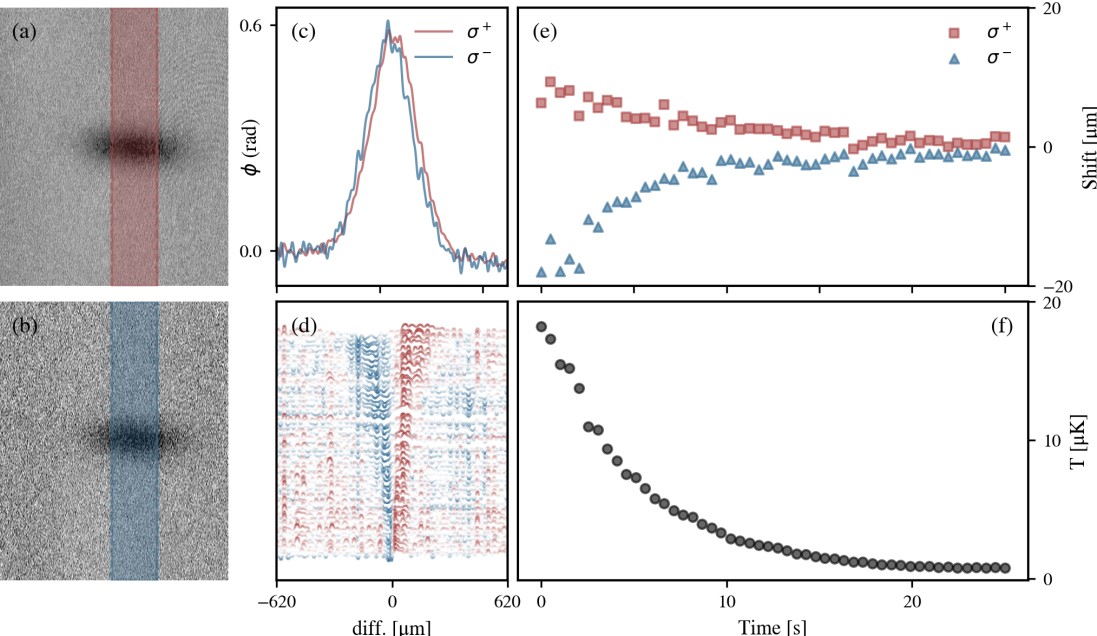

Figure 1: Analysis of the relative shift between the two images corresponding to the opposite circular polarization components of the probe beam. The left-most images show the accumulated phase of **a)** $\sigma_+$ polarized light (red) and **b)** $\sigma_-$ polarized light (blue) at the beginning of the cooling sequence. The shaded regions indicate the integration window used to compute vertical linecuts of the phase profile; image within this region is averaged along the axial direction to yield one-dimensional cross-sections shown in **c)**. The differences between the profiles at various stages of cooling are plotted in **d)**; each line is vertically offset for clarity, and the color encodes the magnitude of the relative shift. The trend clearly indicates a relative shift between two distributions, which diminishes as the temperature decreases, resembling a zipping behavior. Additionally, each distribution is fitted and their relative displacement of the centers is plotted in **e)**, indicating the same pattern. For reference, the corresponding temperatures extracted from the fits to the data are shown in **f)**.

the centers of fitted distributions depending on the time of evaporation, suggesting that the observed shift depends on the temperature. This strongly suggests a physical origin tied to the atom-light coupling, which motivated a detailed theoretical treatment of the polarized probe's interaction with an inhomogeneous magnetic field, presented in the following sections.

## 3 Theoretical framework

To better evaluate the origin of these shifts and quantify the role of magnetic field geometry in spin-resolved imaging of cold atoms, we develop a general model describing the birefringence experienced by circularly polarized light by atoms in spatially varying magnetic fields. This model combines atomic tensor polarizability with local rotations of the spin quantization axis, enabling a precise description of the accumulated phase shift for arbitrary field configurations and probe polarizations.

The description begins with the response of the atomic ensemble to an applied electromagnetic field. This response is captured by the atomic polarizability, which links the local electric field to the induced atomic polarization. The atomic polarizability ultimately determines both the absorption and phase shift experienced by the probe. In spin-polarized gases such as ours, the response is anisotropic and depends on the internal spin state as well as the orientation of the local magnetic field. In this section, we develop a description of this polarizability, beginning from the general susceptibility tensor and reducing it to the case where a linearly polarized probe decomposes into circular components that couple selectively to $\sigma^+$ or $\sigma^-$ transitions.

## 3.1 Polarizability

The propagation of an electromagnetic field through a cloud of cold atoms is governed by the electric displacement field $\boldsymbol{D} = \varepsilon_0 \boldsymbol{E} + \boldsymbol{P}$, where the electric field $\boldsymbol{E}$ is modified by the polarization $\boldsymbol{P}$. In the low-saturation limit, the polarization is linearly related to the electric field as given by the relation

$$\boldsymbol{P} = \varepsilon_0 \overleftrightarrow{\chi} \boldsymbol{E}, \tag{1}$$

where $\overleftrightarrow{\chi}$ is the electric susceptibility tensor. This produces a refractive index $n^2 = 1 + \langle \chi \rangle$, where $\langle \chi \rangle$ denotes the component of the susceptibility tensor projected along the polarization direction of the probe, effectively capturing the optical response experienced by the field. For anisotropic media such as spin-polarized atomic ensembles, the susceptibility must be modeled as a rank two tensor. It is expressed as [22]

$$\overleftrightarrow{\chi} = \rho(\boldsymbol{r}) \frac{\overleftrightarrow{\alpha}}{\varepsilon_0}, \tag{2}$$

where $\rho(\boldsymbol{r})$ is the density of the atoms, and $\overleftrightarrow{\alpha}$ is the polarizability tensor. Here we set the correction term $C$, as defined in [22] to zero, since the real part of $\rho \overleftrightarrow{\alpha}$ is small compared to 1 [23]. In this limit, $\overleftrightarrow{\alpha}$ is given by [24]

$$\overleftrightarrow{\alpha} = \sum_{g,g',e} \frac{i}{\hbar} \frac{1}{\gamma/2 - i\delta} \langle g | \boldsymbol{\mu}_{ge} | e \rangle \langle e | \boldsymbol{\mu}_{eg} | g' \rangle \langle g' | \sigma_{gg} | g \rangle, \tag{3}$$

where the sum is taken over all sublevels of the ground state $g, g'$ that are coupled to an excited state $e$ via the electric-dipole matrix element $\boldsymbol{\mu}_{eg}$. Here, the linewidth $\gamma$ is given by [25]

$$\gamma = \frac{\omega^3 \mu^2}{3\pi \varepsilon_0 \hbar c^3}. \tag{4}$$

For a probe of fixed polarization, each ground Zeeman sublevel couples to only one excited level, permitting adiabatic elimination of the excited states $e$, leaving an effective description that involves only the diagonal ground state populations $\sigma_{gg}$. These are first evaluated in the atomic frame and then rotated in the probe frame $\mathbf{k}$ to yield the relevant coupling strengths. For large probe detunings $|\delta|$ that greatly exceed the excited-state hyperfine splittings, the reduced dipole matrix element $\mu = |\boldsymbol{\mu}_{eg}|$ can be treated as approximately constant across all ground sublevels. Note that it is implicitly assumed that the Zeeman shifts of the ground-state atoms are smaller than the natural linewidth $\gamma$, which is not necessarily the case here.

In the spin-dependent off-axis holography (SOAH) geometry, a linearly polarized probe beam propagates along the $y$-axis and can be decomposed as

$$\hat{u}_0 = \frac{1}{\sqrt{2}} (\hat{u}_{+1} + e^{i\phi} \hat{u}_{-1}), \tag{5}$$

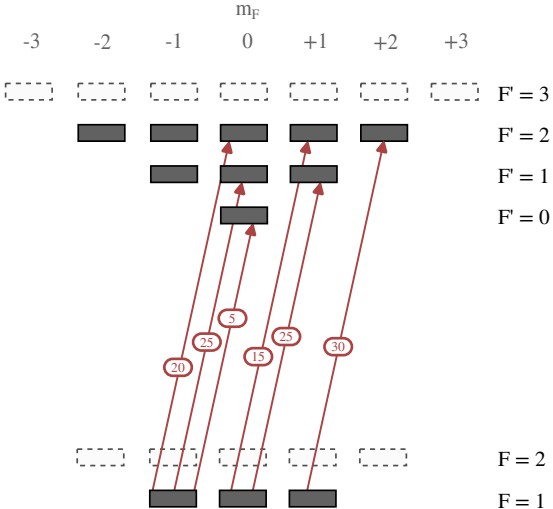

Figure 2: Coupling strengths of the various transitions from the $F = 1$ ground state of sodium with right-handed circular polarized light. All strengths have to be normalized to the strongest transitions, which has a strength of 60. Note that the total coupling from the different magnetic substates to all excited states are ⁵/₆, ⁴/₆, and ³/₆ for $m$ = -1, 0, and +1, respectively. Values taken from [26].

where the circular polarization unit vectors are

$$\hat{u}_{\pm 1} = \frac{1}{\sqrt{2}}(\hat{x} \pm i\hat{z}).$$ (6)

Here, $\phi$ determines the polarization angle in the $xz$-plane.

The holographic imaging scheme separates the left- and right-circular components, allowing for independent treatment of each. For either component, only a single circular dipole transition ($\sigma^+$ or $\sigma^-$) has to be considered. In this case, $\overleftrightarrow{\alpha}$ reduces to a diagonal element ($g' \equiv g$):

$$\overleftrightarrow{\alpha} = \sum_{g,e} \frac{i}{\hbar} \frac{1}{\gamma/2 - i\delta} |\langle g|\boldsymbol{\mu}_{ge}|e\rangle|^2 \langle g|\sigma_{gg}|g\rangle.$$ (7)

Note that the reduction of $\overleftrightarrow{\alpha}$ to a diagonal matrix greatly simplifies the analysis. For sodium atoms in the $F = 1$ hyperfine ground state, the transition strengths for $\sigma^+$ light are proportional to the squares of the Clebsch-Gordan coefficients. These yield total coupling strengths of 5/6, 4/6, and 3/6 for $m = -1, 0, +1$, respectively [26], as shown in Fig. 2.

To account for the local magnetic field orientation, the ground state density matrix is rotated from the field-aligned frame (where atoms occupy the $m = -1$ state) to the probe beam frame (aligned with $y$). The rotation is

$$\tilde{\sigma}_{gg}(\beta) = R_x(\beta)^\dagger \sigma_{gg} R_x(\beta),$$ (8)

in which $\beta$ denotes the angle between the magnetic field and the $y$-axis.

## 3.2 Magnetic field geometry

The magnetic field in a clover-leaf trap can be approximated by [27]

$$B_x = B'x - B''xz,$$ (9)

$$B_y = -B'y - B''yz,$$ (10)

$$B_z = B_0 + B''\left(z^2 - \frac{x^2 + y^2}{2}\right),$$ (11)

Table 1: Experimentally measured geometric coefficients for the cloverleaf trap [21].

| Parameter | Value |
|:---:|:---:|
| $K_c''$ | $1.87 \times 10^{-3}$ G/(A mm$^2$) |
| $K_{ab}''$ | $-1.38 \times 10^{-4}$ G/(A mm$^2$) |
| $K'$ | $3.50 \times 10^{-2}$ G/(A mm) |
| $K_c$ | $1.03$ G/A |
| $K_{ab}$ | $-1.02$ G/A |

where $B_0$ is the field minimum, $B'$ the total gradient in the radial direction, and $B''$ the curvature in the axial direction. These parameters are determined by the currents $I_g$, $I_c$, and $I_{ab}$ through the gradient (cloverleaf), curvature, and antibias coils, respectively. The clover-leaf trap, such as the one used in our experiment, is described in greater detail in [20], with the relevant magnetic fields described by

$$B_0 = I_c K_c + I_{ab} K_{ab}, \tag{12}$$

$$B' = I_g K', \tag{13}$$

$$B'' = I_c K_c'' + I_{ab} K_{ab}''. \tag{14}$$

The geometric factors $K_c, K_{ab}, K', K_c'', K_{ab}''$ depend on the geometry of the coils and are experimentally determined with the magnetic field measurements and are listed in Table 1. The current through the clover-leaf coils is $I_g = 329$ A, the current through the curvature coils is $I_c = 200$ A, and the current through the antibias coils is $I_{ab} = 200$ A.

In the radial direction, the clover-leaf trap potential transitions from harmonic to linear when the thermal energy equals the Zeeman energy, i.e. when $k_B T = \mu B_0$. For our operating conditions, with $B_0 \approx 2$ G, this crossover occurs at $T \approx 70$ $\mu$K. Since the highest temperatures reached in our experiment are on the order of 20 $\mu$K, the radial confinement is already well within the harmonic regime. Along the axial direction, the trap remains harmonic at all temperatures of interest.

The angle $\beta$ between the local magnetic field vector and the probe propagation direction ($y$) is computed as

$$\cos\beta = \frac{B_y}{\sqrt{B_x^2 + B_y^2 + B_z^2}}. \tag{15}$$

This angle varies across the cloud, resulting in spatial modulation of the local quantization axis.

Figure 3 illustrates the variation of $\beta$ in the $xz$-plane for a thermal sodium cloud at $T = 30$ $\mu$K, showing a tilt of the magnetic field away from the $y$-axis. This induces birefringence predominantly along the radial direction.

## 3.3 Rotation matrices

The atomic polarizability under a magnetic field requires alignment of the quantization axis with the propagation direction of the incident radiation. The laboratory frame spherical angles $\theta$ (polar) and $\phi$ (azimuthal) are defined via the magnetic field using the relation

$$\cos\theta = \frac{B_y}{B_{\text{tot}}}, \qquad \tan\phi = \frac{B_z}{B_x}, \tag{16}$$

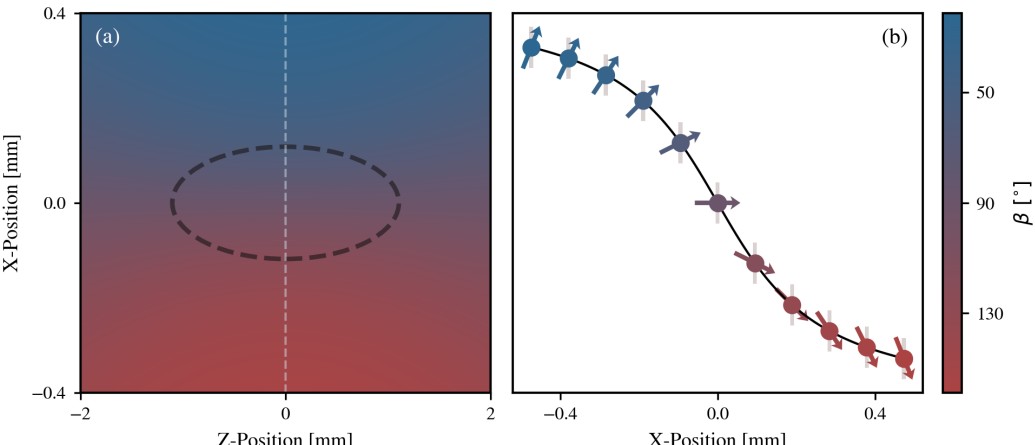

Figure 3: Orientation of the magnetic field characterized by the angle $\beta$ relative to the $y$-direction in the $xz$-plane, at $y = 0$ (color). The dashed ellipse represents the spatial extent of a thermal atomic cloud at $T = 30\ \mu$K in *in-situ* conditions. Here, $\sigma_z = \sqrt{k_B T/m\omega_z^2} \approx 1.1$ mm and $\sigma_r = \sqrt{k_B T/m\omega_\rho^2} \approx 0.12$ mm, with $m$ in this case denoting the mass of sodium atoms, and $\omega_z$, $\omega_\rho$ the trapping frequency in axial and radial direction, respectively. The arrows in the right-hand plot indicate the direction of the quantization axis relative to the $y$-direction - the direction of the propagation of light - along the center of the trap ($y = 0$, $z = 0$).

where $B_{\text{tot}} = \sqrt{B_x^2 + B_y^2 + B_z^2}$. To rotate the field vector on the $y$ axis, a rotation by $-\phi$ about $\hat{\mathbf{y}}$ is applied followed by a rotation by $-\theta$ about $\hat{\mathbf{z}}$. The combined transformation in Euler-angle convention[2] is

$$D(\alpha, \beta, \gamma) = D_y(0)D_z(-\theta)D_y(-\phi), \tag{17}$$

with $(\alpha, \beta, \gamma) = (-\phi, -\theta, 0)$. For the total angular momentum quantum number $F = 1$, the general Wigner rotation D-matrix takes the form

$$D_{m'm}^{(1)}(\alpha, \beta, \gamma) = e^{im'\gamma} d_{m'm}^{(1)}(\beta) e^{im\alpha}, \tag{18}$$

where $m, m'$ can take values $\{-1, 0, 1\}$, and $d_{m,m'}^{(j)}(\beta)$ is the Wigner small $d$-matrix, which represents the rotation of angular momentum states with total angular momentum $j$ around the $y$-axis by the angle $\beta$. This leads to a 3-by-3 matrix, which after introducing the relevant angles $\theta, \phi$ becomes [28]

$$D_{m'm}^{(1)}(-\theta, -\phi, 0) = \begin{bmatrix} \frac{1}{2}(1+\cos\theta)e^{i\phi} & \frac{1}{\sqrt{2}}\sin\theta & \frac{1}{2}(1-\cos\theta)e^{-i\phi} \\ -\frac{1}{\sqrt{2}}\sin\theta e^{i\phi} & \cos\theta & \frac{1}{\sqrt{2}}\sin\theta e^{-i\phi} \\ \frac{1}{2}(1-\cos\theta)e^{i\phi} & -\frac{1}{\sqrt{2}}\sin\theta & \frac{1}{2}(1+\cos\theta)e^{-i\phi} \end{bmatrix}, \tag{19}$$

with the upper-left element representing $m' = -1$, $m = -1$. Applying this transformation to the density matrix yields

$$\tilde{\sigma}_{gg} = D(\alpha, \beta, \gamma)^\dagger \sigma_{gg} D(\alpha, \beta, \gamma), \tag{20}$$

---

[2]Euler angles $(\alpha, \beta, \gamma)$ define rotations in the Wigner $D$-matrix formalism. Note that $\beta$ above (deviation from the $y$ axis) differs from the Euler angle $\beta$.

where † denotes the Hermitian conjugate, and an initial nonrotated density matrix $\sigma_{gg}$ is given by

$$\sigma_{gg} = \begin{bmatrix} 1 & 0 & 0 \\ 0 & 0 & 0 \\ 0 & 0 & 0 \end{bmatrix}, \tag{21}$$

where the population is initially in the $m = -1$ state. Equation (20) then becomes:

$$\tilde{\sigma}_{gg} = \begin{bmatrix} \cos^4(\theta/2) & \frac{1}{\sqrt{2}}e^{-i\phi}\cos^2(\theta/2)\sin\theta & \frac{1}{4}e^{-2i\phi}\sin^2\theta \\ \frac{1}{\sqrt{2}}e^{+i\phi}\cos^2(\theta/2)\sin\theta & \frac{1}{2}\sin^2\theta & \frac{1}{\sqrt{2}}e^{+i\phi}\sin^2(\theta/2)\sin\theta \\ \frac{1}{4}e^{+2i\phi}\sin^2\theta & \frac{1}{\sqrt{2}}e^{-i\phi}\sin^2(\theta/2)\sin\theta & \sin^4(\theta/2) \end{bmatrix}. \tag{22}$$

Retaining only diagonal elements yields the population fractions

$$\Pi_{-1} = \cos^4\left(\tfrac{\theta}{2}\right), \qquad \Pi_0 = \tfrac{1}{2}\sin^2\theta, \qquad \Pi_{+1} = \sin^4\left(\tfrac{\theta}{2}\right), \tag{23}$$

which satisfy $\sum_m \Pi_m = 1$. For $\theta = 0$, the entire population occupies $m = -1$; for $\theta = \pi$, $m = +1$; and for $\theta = \pi/2$, one finds $\Pi_{\pm 1} = 1/4$ and $\Pi_0 = 1/2$.

## 3.4 Accumulated phase and column densities

Determination of the accumulated phase shift experienced by probe light traversing the atomic ensemble is achieved by integrating the local phase increment along the propagation axis. The phase profile for a beam with some polarization $q$, $\phi_q(x,z)$ is expressed as [19, 29]

$$\phi_q(x,z) = \frac{ik}{2\hbar\varepsilon_0} \sum_{m,F'} \frac{C_{mF'}^q}{(\gamma/2) - i\,\delta_{F'}} \Pi_m(\beta) \int \rho_m(x,y,z)\,dy. \tag{24}$$

Here, $\rho_m(x,y,z)$ denotes the spatial density of the atomic gas of a magnetic substate $m$, $k$ is the optical wavenumber, $\gamma$ represents the natural linewidth and $\delta_{F'}$ is the detuning relative to the excited hyperfine manifold $F'$. The Clebsch-Gordan coefficients $C_{m,F'}^q$ quantify the dipole coupling strengths between ground sublevel $m$ and the excited state $F'$, and depend on the polarization of the probe. The projection factor $\Pi_m(\beta)$ encodes the sublevel population distribution as a function of the rotation angle $\beta$.

In the far-detuned regime ($|\delta_{F'}| \gg \gamma, |\delta_{F'} - \delta_{F''}|$), the sum over $F'$ may be approximated by an effective detuning $\delta$ and total coupling for a given polarization $C_m^{q,(\text{tot})} = \sum_{F'} C_{mF'}^q$, which yields

$$\phi_q(x,z) \approx \frac{ik}{2\hbar\varepsilon_0} \frac{1}{(\gamma/2) - i\delta} \sum_m C_m^{q,(\text{tot})} \Pi_m(\beta) \int \rho_m(x,y,z)\,dy. \tag{25}$$

For a thermal cloud characterized by a three-dimensional, radially symmetric Gaussian distribution,

$$\rho(x,y,z) = \rho_0 \exp\left[-\frac{x^2 + y^2}{2\sigma_r^2} - \frac{z^2}{2\sigma_z^2}\right], \tag{26}$$

with radial and axial widths $\sigma_r, \sigma_z$, the spatial dependence of $\Pi_m(\beta)$ induces an asymmetric phase profile for circularly polarized light. Specifically, for right- versus left-handed circular polarization, the phase contour is displaced oppositely along the radial axis. The simulation in Fig. 4 illustrates this effect for a cloud of $3 \times 10^8$ atoms at $30\,\mu K$, exhibiting a radial shift of approximately $\pm 40\,\mu m$ relative to the centroid of the phase distribution at $\beta = \pi$. Although an axial shift is, in principle, also expected, it depends on the magnetic-field gradient along the axial direction, which is negligible near the trap center. As a result, no significant axial displacement is observed in the simulations or in the experiment.

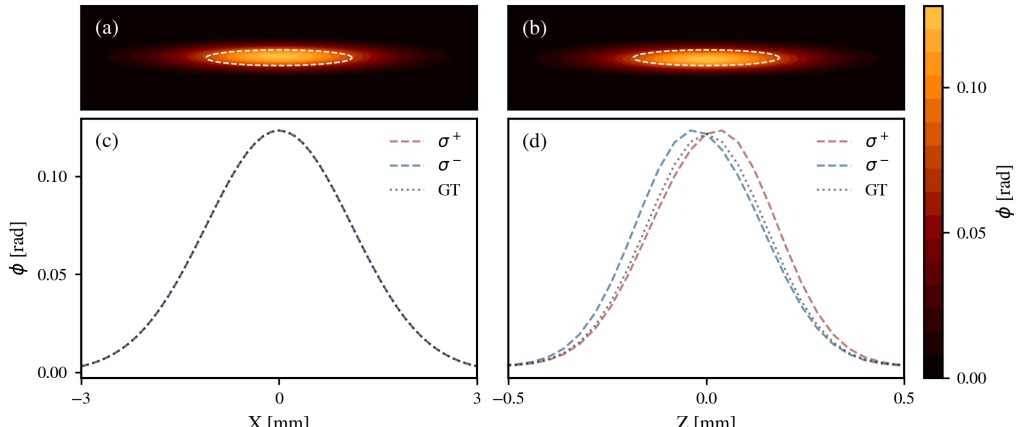

Figure 4: Simulated phase delay contours for **a)** $\sigma_+$ and **b)** $\sigma_-$ polarizations showing an apparent displacement of the phase centroid in a thermal atomic cloud of $3 \times 10^8$ atoms at $30\,\mu$K. The dashed white line indicates its Gaussian widths. **c)** Vertical and **d)** horizontal line cuts through the center of the clouds highlight an apparent radial shift of about $\pm 40\,\mu$m for $\sigma^\pm$-polarized light, relative to the center of the ground truth (GT) phase delay for $\beta = \pi$. Note the different x-axis scales in the plots in **c)** and **d)** due to the cigar-shaped geometry of the cloud. The vertical cut is scaled for better visualization of the displacement.

## 4 Comparison to the experimental data

For each fitted set of physical parameters from the experimental data, synthetic data are simulated as described in the sections above, and centroids are determined through the same fitting procedure as the experimental data. The results are presented in Figure 5. For high temperatures above $T > 2\,\mu$K, the simulation reproduces the measured shift well, pointing to the magnetic field inhomogeneity as the likely cause. However, for temperatures below $2\,\mu$K, a residual offset of approximately $1\,\mu$m persists (Fig. 5c). The magnitude of the excess shift appears to be independent of magnetic gradients, as a shift of identical magnitude was observed in an all-optical dipole trap experiment, where $B(\mathbf{r})$ was uniform [19].

Many imaging protocols extract physical information by matching two related images pixel by pixel and forming a local ratio or difference. If the two images were taken with different probe polarizations, or the magnetic landscape has changed between the images, such a shift needs to be corrected for. In our case the field in the center of the magnetic trap is almost a pure linear gradient, which results in only a uniform lateral displacement of the two images, which can be fixed in post-processing. In a more complex magnetic environment, the phase error incurred due to the inhomogeneity of the field will go beyond a simple shift and instead lead to aberrations of the images, where the aberrations will be different for the two polarizations.

In the case of digital holography schemes, where full field information is obtained, such aberrations can be corrected by multiplying the images in Fourier space with a synthetic phase mask $\Delta\Phi_{\text{field}}(x, z)$ [18, 30–34]

$$I_{real}(x, z) = \mathcal{F}^{-1}\left\{\exp\left[-i\Delta\Phi_{\text{field}}(x, z)\right]\tilde{I}(k_x, k_z)\right\}, \tag{27}$$

where $\tilde{I}(k_x, k_z)$ is the spatial Fourier transform of the measured intensity field and $(k_x, k_z)$ the spatial frequencies. For small angles $\beta$ (near the center of the trap) where $\Delta\Phi_{\text{field}}(x, z)$ is approximately linear, the correction can be performed by applying a 2D linear gradient in the Fourier space,

$$\Delta\Phi_{\text{field}}(x, z) \approx k_x x_0 + k_z z_0. \tag{28}$$

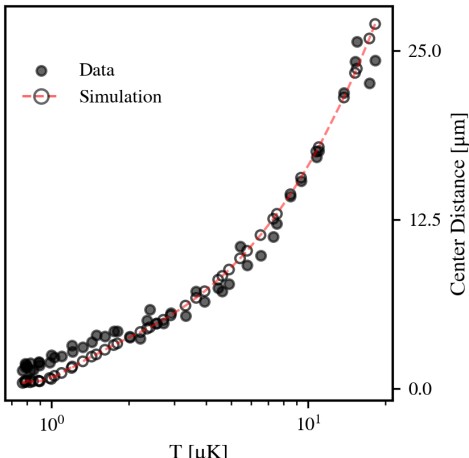

Figure 5: Displacement between the centers of the two circular polarization components as a function of temperature. For each image taken during cooling, the temperature is extracted from fits to the data, and the corresponding displacement is computed using the model described above. At high temperatures, the model predicts the behavior well, but it seems not to fully account for the shift at lower temperatures. The solid black dots show the data; empty dots with a dashed red guide show the simulation.

Optimal parameters $x_0$ and $z_0$ can in principle be derived from numerical simulations, but since it may sometimes not be possible to precisely know the magnetic field in detail, it is often sufficient to find these factors by minimizing the mean square difference between the centers of the fits to the reconstructed density profiles.

As mentioned above, the same correction strategy also works for richer magnetic landscapes, such as tailored lattices of magnetic microtraps [35–37], atomic chips [38–40] as well as the kilogauss-scale traps used for Feshbach resonance tuning [41–43]. Although these traps can strongly distort images and introduce higher-order aberrations, if the magnetic field is known, $\Delta\Phi_{\text{field}}(x,z)$ is computed precisely in terms of the angle $\beta$ given in Eq. 15,

$$\Delta\Phi_{\text{field}}(x,z) = \int \mathrm{d}y \; \kappa_{\text{eff}} \sin\beta(x,y,z) \left(1 - \tfrac{1}{2}\sin^2\beta(x,y,z)\right), \tag{29}$$

where the single scale factor $\kappa_{\text{eff}}$ is fixed once from a reference shot. This method thus allows aberration correction for any known magnetic field.

## 5 Conclusion

We have introduced a compact three-dimensional framework that links the local tilt of the magnetic field to the phase accumulated by the circular components of an off-axis probe. By combining tensor polarizability with field-frame rotations, the model predicts both phase and spin density transformations, provided $\mathbf{B}(x,y,z)$ is known. The theoretical predictions follow experimental SOAH data down to a few $\mu K$, and reveal a systematic displacement of roughly $\sim 1\,\mu\text{m}$ that persists even in a uniform dipole trap. For small clouds and for $\beta \ll 1$, this factor can be seen as roughly linear, allowing for a fast correction even when the magnetic fields are not precisely known. When the magnetic field is known exactly, this method allows for higher-order aberration corrections down to the diffraction limit.

Our model, combined with the Fourier-space phase mask, provides a universal correction for polarization-dependent aberrations introduced by inhomogeneous magnetic fields. This approach enables reliable spin-resolved imaging in complex magnetic landscapes and improves spatial resolution in studies of quantum gases subjected to arbitrary field geometries.

## Acknowledgments

The authors thank A. Mosk for useful discussions.

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
