# Peer review of "Polarization-Sensitive Imaging in Magnetic Environments"

_SciPost Physics Core, doi:SciPost Phys. Core 8, 076 (2025)_

## Round 1 · Referee Report · Anonymous (Referee 1) · 2025-8-25

Report

The paper "Spin-dependent Off-Axis Holography in Magnetic Environments" examines an aspect of the imaging process of ultracold atomic clouds. It is thus an addition to previous publications by the authors on the use of "Off-Axis Holography". In such previous work, the authors had noticed an offset between images of the same cloud taken with two different circular polarizations.
The main aim of the present paper is to analyse the effect of a magnetic environment (a trap) on the dispersion of light with $\sigma^+$ or $\sigma^-$ polarization by an ultracold cloud. The authors find that dispersion under such conditions indeed leads to a significant offset of the resultig images, confirming largely the experimental observation.
The paper is well written and can be followed relatively easily. However, I had to make considerable detours to Refs 18 and 19 to understand the SOAH imaging technique. Maybe a bit more information could be given here, such that the manuscript is somewhat more self-contained.
I was also a bit surprised to see the same data presented twice in Figs 1 and 5. Maybe the first figure could be reduced just to show the main effect and thus avoid revisiting all the data.
Finally, I am puzzled by the title. In my understanding, the analysis is very general and shows the different displacements/distortions of $\sigma^+$ and $\sigma^-$ light by ultracold atoms. This happens in many experiments, but is typically not noticed, since only one of these polarization orientations is used. Here, the SOAH technique is used to observe this effect, but that is not the main result. I would therefore suggest to use a tite which directly reflects the main result.

Requested changes

  1. Please add some more information on SOAH, such that the manuscript is somewhat more self-contained.
  2. I suggest to reduce the first figure to show the main effect and thus avoid revisiting all the data.
  3. I suggest changing the title to reflect the main result.

Recommendation

Publish (easily meets expectations and criteria for this Journal; among top 50%)

  • validity: top
  • significance: good
  • originality: high
  • clarity: top
  • formatting: excellent
  • grammar: perfect

Author:  Peter van der Straten  on 2025-09-10  [id 5803]

(in reply to Report 1 on 2025-08-25)

Rebuttal to the first referee report

We thank the referee for carefully reading the manuscript, his appraisal and providing us with his suggestions. We agree with his comments and have adjusted the manuscript accordingly (see enclosed manuscript with the changes shown in red and the added sections shown in blue).

Our response to the comments is as follows:

  1. Please add some more information on SOAH, such that the manuscript is somewhat more self-contained.

We have been very short on the description of the technique, since we are using the same setup as in a previous paper. However, we understand that adding more information makes the article more self-contained. We have added a full paragraph describing the technique at the start of Sec. II.

  1. I suggest to reduce the first figure to show the main effect and thus avoid revisiting all the data.

We have reduced Fig. 5 considerably, in order not to revisit the data. In Fig. 1 the temperature is important, since it provides a clue to the cause of the effect. But, it is not necessary to repeat the information in Fig. 5.

  1. I suggest changing the title to reflect the main result.

We thank the referee for this suggestion. Indeed, the current title does not reflect the main result. Thus we propose to change the title to "Polarization-sensitive imaging in magnetic environments".

On behalf of the authors, Peter van der Straten

Attachment:

Polarization_sensitive_imaging_in_magnetic_environments__Dif_A9Uk0J0.pdf

---

## Round 1 · Referee Report · Anonymous (Referee 2) · 2025-9-5

Strengths

The three-dimensional modelling of the magnetic field orientation is very good.

A thorough theoretical modelling revealing sophisticated experimental signatures.

Weaknesses

The paper is very technical. Since it has been submitted to the "Physics Core" section, which aims at an audience across all physics, I would suggest to improve the introductory paragraphs (II and III. A) by some more elaborate wording that could make the text more comprehensible/appealing to a non-expert.

Report

The paper gives a very detailed account of the interaction of ultracold sodium atoms with light fields of different polarizations. The authors refine their previous work on spin-dependent off-axis holography and present a thorough theoretical modelling that can account for many of the observed features, including the apparent positional shift of the cloud in the images of orthogonal polarizations. The temperature dependence of this effect is striking and the authors can explain this very well with their model.

The authors work in the framework of the harmonic approximation of the Ioffe-Pritchard trapping potential. Whether this is valid or not at their initial temperatures, in particualr for the outer regions of the trapped atom cloud, cannot be assessed from the paper. It would be desireable to have a comment to about this.

I failed to find the information about the optical density (OD) at which these experiments have been performed. Does this affect the interpretation of the data, in particular, since this will vary when changing the temperature. What is the contribution of OD gradients to the images?

How sensitive are the data and interpretation to the detuning of the probe light, which seems to be chosen at a fixed value for the manuscript. How was this value selected?

Requested changes

see above.

Recommendation

Publish (easily meets expectations and criteria for this Journal; among top 50%)

  • validity: high
  • significance: high
  • originality: high
  • clarity: ok
  • formatting: excellent
  • grammar: excellent

Author:  Peter van der Straten  on 2025-09-10  [id 5802]

(in reply to Report 2 on 2025-09-05)

Rebuttal to the second referee report

We thank the referee for carefully reading the manuscript, his appraisal and providing us with his suggestions. Here we provide our reaction to the points raised by the referee. For the changes, please see the enclosed manuscript with the changes shown in red and the added sections in blue.

  1. The paper is very technical.

We have added new introductionary material to both Sec. II and III.A, as suggested by the referee. Here we describe in a more descriptive way the contents of the two sections to a wider audience.

  1. The authors work in the framework of the harmonic approximation of the Ioffe-Pritchard trapping potential.

We have added a paragraph to Sec. III.B describing that our trap is in the harmonic regime for the temperatures that we use.

  1. I failed to find the information about the optical density (OD) at which these experiments have been performed.

We have added a paragraph to Sec. II starting with "In our implementation .." that discusses the choices that we have made regarding the detuning of the light, the effect of the OD, and the phase shifts.

On behalf of the authors, Peter van der Straten

Attachment:

Polarization_sensitive_imaging_in_magnetic_environments__Dif_yzc8nWj.pdf

---

## Round 2 · Author Response

We have carefully read the reports by the referees. We are pleased that the referee reports have been positive and agree with their suggestions. We have implemented the changes, as indicated in our response to the referees. We also included a manuscript highlighting the changes, where the corrections are indicated in red and the added sections can be seen in blue.
On behalf of the authors, Peter van der Straten

---

## Round 2 · List of Changes



---

## Editorial Decision

published